# The Role of Ectodysplasin A on the Ocular Surface Homeostasis

**DOI:** 10.3390/ijms232415700

**Published:** 2022-12-10

**Authors:** Shangkun Ou, Mani Vimalin Jeyalatha, Yi Mao, Junqi Wang, Chao Chen, Minjie Zhang, Xiaodong Liu, Minghui Liang, Sijie Lin, Yiming Wu, Yixuan Li, Wei Li

**Affiliations:** 1Eye Institute of Xiamen University and Affiliated Xiamen Eye Center, School of Medicine, Xiamen University, Xiamen 361000, China; 2Fujian Provincial Key Laboratory of Corneal & Ocular Surface Diseases, Xiamen 361000, China; 3Fujian Provincial Key Laboratory of Ophthalmology and Visual Science, Xiamen 361000, China; 4Department of Ophthalmology, Graduate School of Medicine, Osaka 5650871, Japan; 5Xiang’an Hospital of Xiamen University, School of Medicine, Xiamen University, Xiamen 361000, China

**Keywords:** ectodysplasin A, ocular surface, homeostasis

## Abstract

Ectodysplasin A (EDA), a ligand of the TNF family, plays an important role in maintaining the homeostasis of the ocular surface. EDA is necessary for the development of the meibomian gland, the lacrimal gland, as well as the proliferation and barrier function of the corneal epithelium. The mutation of EDA can induce the destruction of the ocular surface resulting in keratopathy, abnormality of the meibomian gland and maturation of the lacrimal gland. Experimental animal studies showed that a prenatal ultrasound-guided intra-amniotic injection or postnatal intravenous administration of soluble recombinant EDA protein can efficiently prevent the development of ocular surface abnormalities in EDA mutant animals. Furthermore, local application of EDA could restore the damaged ocular surface to some extent. Hence, a recombinant EDA-based therapy may serve as a novel paradigm to treat ocular surface disorders, such as meibomian gland dysfunction and corneal epithelium abnormalities.

## 1. Introduction

Ectodysplasin A (EDA), encoded by the EDA gene positioned in the X chromosome, is a member of the tumor necrosis factor (TNF) superfamily that contribute to cell death, proliferation or differentiation [1]. However, EDA is a unique member of the TNF ligand because of its limited sequence homology to other TNF-like molecules except for the conserved TNF motif [2]. The EDA gene governs the morphogenesis of various ectodermal organs such as the teeth, hairs, and mammary glands during prenatal development [3]. A literature survey revealed that, among several signaling pathways, the EDA pathway was the first pathway to be utilized for the stimulation of tooth modifications. EDA gene mutations are widely studied in X-linked hypohidrotic ectodermal dysplasia (XLHED) and anhidrotic/hypohidrotic ectodermal dysplasia (HED), which is the most common genetic disorder of ectodermal development in humans resulting in hypotrychosis, hypodontia, heat intolerance, dry skin and dry eyes, the susceptibility to airway infections and crusting of various secretions. Here, we review the signaling pathways involved in EDA, its role in the morphogenesis of the ocular surface and the emergence of recombinant EDA as a bioactive compound for the management of ocular surface disorders. 

## 2. EDA/EDA Receptors System

EDA precursor protein is a transmembrane protein, which contains a short intracellular domain, a transmembrane domain and an extracellular domain. The extracellular domain consists of three functionally important regions of EDA: a furring protease recognition sequence responsible for proteolytic processing of EDA, a collagen-like domain and a C-terminal TNF homology domain responsible for receptor binding [2,4]. The ectoderm-derived epithelial cells express the EDAR, and a hinderance to the EDA-EDAR signaling pathway leads to genetic disorders such as anhidrotic ectodermal dysplasia. The EDA transcript undergoes complicated splicing events generating various splice variants, among which EDA-A1 (391 amino acid) and EDA-A2 (389 amino acid) are the common functional variants. They differ by an insertion of just two amino acid residues (Glu308 and Val309) in the TNF domain [5]. Despite the high sequence homology, EDA-A1 and EDA-A2 specifically bind to two different receptors, the EDAR and X-linked EDA-A2 receptor (XEDAR), respectively [6]. Being splice variants, the function and signaling proteins of the EDA A1-EDAR pathway and EDA-A2-XEDAR pathway are distinct [7]. EDA-A1-EDAR affects the development of skin appendages, including hair, teeth, sweat glands, meibomian glands and preputial glands [8], whereas EDA-A2-XEDAR is a p53-induced gene with no obvious implications in ectodermal appendage development [7]. 

EDA can regulate the NF-κB, Wnt, Shh, BMP and lymphotoxin-β (LTβ) pathways spatio-temporally in the organogenesis and the maintenance of the ectodermal organ (Figure 1).

## 3. EDA-EDAR-Dependent Signaling Pathways

### 3.1. NF-κB Signaling Pathway

Among various signaling pathways involved in prenatal organogenesis, the NF-κB pathway plays a major role in the development of most ectodermal organs. Many studies on EDA mutant Tabby mice have proved that EDA-EDAR regulates the NF-κB signaling pathway. 

Analogous with other TNF family receptors, EDAR trimerizes upon binding of the EDA-A1 ligand [9]. However, EDAR cannot bind to any of the TNF-associated factors (TRAFs) directly; it requires a special adapter protein EDAR-associated death domain (EDARADD), to recruit tumor necrosis factor receptor-associated factors (TRAFs) for the activation of the downstream NF-κB signaling pathway [10,11] (Figure 1). As reported, among the six TRAFs, only TRAF6 is involved in the activation of the EDA-A1-mediated NF-κB signaling pathway [12,13]. Similarly, the XEDAR receptor undergoes ligand-mediated trimerization [9,14] and can recruit TRAF3 and TRAF6 [12], thereby activating the NF-κB signaling pathway [15]. When the TRAFs bind to the EDARADD, it activates the IKK complex that consists of NEMO, IKKα and IKKβ. The IKK complex induces the phosphorylation of Iκb. The phosphorylated Iκb is degraded and then releases NF-κB. NF-κB translocates into the nucleus and activates the target genes [12,13]. If the X chromosome carried the mutated EDA gene, NF-κB dysfunction would be evident through phenotypic abnormalities in the development of the mammary gland [16], teeth [17], hair follicle [18] and meibomian gland [19]. 

### 3.2. Wnt/β-Catenin Signaling Pathway

Together with the NF-κB pathway, the Wnt/β-catenin signaling pathway is critical as they crosstalk bidirectionally to initiate the morphogenesis of hair follicles [20], teeth [21], the meibomian gland [22] and mammary gland [23]. Zhang et al., using a hair follicle induction model, showed that the Wnt and Edar signaling pathways are interdependent when inducing the formation of the primary hair follicle placod and the Wnt directly targets Edar [20]. Similarly, Wang et al. (2020) found that EDAR stimulated the Wnt/β-catenin signaling, promoting tumor cell proliferation in colorectal cancer. The gene expression levels associated with Wnt/β-catenin signaling were upregulated in high EDA samples, while β-catenin expression was significantly downregulated when EDAR was silenced [24], highlighting the importance of EDA-EDAR in the activation of the Wnt/β-catenin signaling pathway during development and tumorigenesis.

### 3.3. BMP Signaling Pathway

Most of the biological processes such as tooth morphogenesis, hair placode formation and hair follicle patterning depend on the interaction between the Eda-Edar and the BMP signaling pathways. An EDAR-BMP activation–inhibition phenomenon was introduced by Mou et al., in which the upregulated BMPs inhibit the EDAR expression during the determination of the follicles’ fate [25]. 

### 3.4. c-Jun N-Terminal Kinase Signaling Pathway

In the process of ectodermal differentiation, EDA-EDAR can activate the c-Jun N-terminal kinase (JNK) signaling pathway, depending especially on the EDA-A2-XEDAR/NF-κB [15,26,27]. A recent study showed that hepatic EDA expression promotes JNK activation and is involved in the obesity-induced insulin resistance in skeletal muscle [28]. Studies showed that the cytoplasmic domain of EDAR resembles the death domains, and mediates the JNK and cell death pathways contributing to pathological phenotypes of anhidrotic ectodermal dysplasia [29]. 

As well as the above signaling pathways, the EDA-EDAR system can activate the sonic hedgehog (Shh) signaling [30] and upregulate FGF20 [31] and EGF [32,33].

## 4. Function of EDA in Physiology and Pathology

EDA is expressed in various organs and tissues, including the heart, kidney, pancreas, brain, lung, liver, skeletal muscle, teeth, as well as the skin during both embryonic development and adulthood [34]. Ever since its discovery in 1996 by D. Schlessinger, numerous studies have determined the role of EDA in the development of ectodermal structures such as the teeth, hair and several exocrine glands including the sweat, mammary and meibomian glands [8,35]. The function of EDA in health and disease is summarized in Table 1. Recently, the expression of the EDA/EDAR receptor system has been extensively studied in the ocular surface as it is a regulator of the ectodermal organs. 

A clinical phenotype associated with EDA gene mutation is X-linked hypohidrotic ectodermal dysplasia (XLHED), also named as anhidrotic/hypohidrotic ectodermal dysplasia (HED), which is the most common genetic disorder of ectodermal development in humans resulting in hypotrychosis, hypodontia, heat intolerance, dry skin, susceptibility to airways infections and crusting of various secretions [36]. In the ocular surface, the abnormal expression of EDA mostly resulted in dry eye disease; that is, the pathology of the lacrimal functional unit (lacrimal gland, cornea, conjunctiva, meibomian glands and so on).

## 5. The Homeostasis of Ocular Surface

The ocular surface is a complicated system, constituting the cornea, conjunctiva, meibomian glands, lacrimal glands and the neural network, which complement each other in maintaining the ocular surface homeostasis [54]. The cornea is the transparent and avascular tissue that serves as a mechanical barrier and refractive surface of the eye. In addition to the tear film, the corneal epithelium is the outermost layer constantly exposed to the external environment. Conjunctiva plays an important role in protecting the eye by producing mucin and the presence of immune cells [55]. The conjunctival epithelium acts as a barrier similar to the corneal epithelium. The lacrimal gland renders lubrication and protects the ocular surface by the secretion of tears consisting of water, electrolytes, lipocalin, lactoferrin and mucus. The function of the lacrimal glands is also necessary for the homeostasis of normal vision [56]. The meibomian glands are the largest sebaceous glands that secrete various lipids including cholesterol, cholesterol esters, wax esters, triglycerides, phospholipids, free cholesterol and free fatty acids. The meibum and aqueous tears make up the stratified structure of the tear film. During the blink reflex, the meibum, aqueous and mucin mix to form the tear film on the ocular surface [57]. The homeostasis of the ocular surface plays an important role in maintaining the health of the eye. The destruction of the homeostasis of the ocular surface results in a variety of diseases, such as meibomian gland dysfunction, corneal epithelium abnormalities, dry eye disease, and etc.

## 6. The Role of EDA in the Development of Ocular Surface

Most of the ocular surface tissues such as the meibomian gland, lacrimal gland and corneal epithelium, originate from the ectoderm [58]. As previously discussed, EDA is involved in the development of several ectodermal organs, including the teeth, hair and mammary glands [59]. Patients with defective EDA are also reported to have photophobia and a reduction in the lacrimal function [60,61,62,63]. Kaercher et al. also observed that these patients presented with alterations in the meibomian glands irrespective of age, and corneal changes in some older patients. Other abnormalities, such as conjunctivitis, lacrimation and dry eye progressed gradually with age in these patients [64]. Similarly, in the naturally occurring animal model of XLHED, the integrity of the cornea, function of lacrimation and formation of the meibomian gland was destroyed [65,66]. The *Tabby* mice, a mice model generated by debilitating the EDA gene, also showed similar clinical characteristics seen in XLHED [67,68]. The mutation of EDA was elucidated to be the major cause of alternation of the meibomian gland in XLHED [67,68]. A further study found that the EDA-DKK4-Lrp6 axis plays a crucial role in the formation of the meibomian gland, and that EDA directly activates the major Wnt pathway modulator Dickkopf-4 (Dkk4) and its receptor Lrp6 during the meibomian gland’s induction [22].

During embryonic development of the lacrimal gland (LG) in mice, the EDA pathway is found to be active in both basal and supra-basal cell layers of the epithelial compartment [52]. However, Eda activity gradually decreased as development proceeded, and there were only a few positive cells in the lacrimal gland acinar domain of the 13-week-old mice [52]. The LG ductal and acinar compartment formation is not affected by the EDA pathway, while EDA is necessary for the terminal differentiation of LG cells and the secretory function of LG during development [52]. Compared to the wild-type mice, the terminal differentiation of cells was found to be altered in all of the LG compartments of EDA^−/−^ mice. Interestingly, the blinking rate remained consistently higher even in one-year-old EDA^−/−^ mice, indicating a long-term physiological defect of the ocular surface in EDA^−/−^ mutants. 

In addition to the *Tabby* mice, Takashi Kuramoto et al. generated an swh/swh rat model by inducing mutation of the Edar-associated death domain (Edaradd) gene, which showed a similar phenotype of meibomian gland and other ocular surface abnormalities in *Tabby* mice [69]. Collectively, the deficiency of EDA contributed to the deformation of the meibomian gland and the immaturity of the lacrimal gland. 

## 7. The Role of EDA in Ocular Surface Homeostasis

As discussed previously, most of the EDA in the ocular surface is contributed by the meibomian gland in the adult stage [32,40], while the LG, corneal and conjunctiva epithelium weakly express EDA [32,40]. However, the EDAR is highly expressed in the cornea, meibomian gland, lacrimal gland and conjunctiva [33] (Figure 2 and Figure 3, Table 2).

### 7.1. Meibomian Gland 

Meibomian gland dysfunction (MGD), a chronic abnormality, could induce dry eye, which affects the health and well-being of millions of people, with terminal obstruction and/or glandular secretion changes [72]. Mutation of the EDA gene induces the abnormal development of the meibomian gland in XLHED patients [73] and animal models of dog [66], mice [53] and rat [69]. We further confirmed that most of the EDA was contributed by the meibomian gland in the ocular surface [33]. The meibomian gland secretes EDA protein to the ocular surface, which in turn contributed to the health of the corneal and conjunctiva [33]. In our previous study, we found that the production of EDA in tears was dramatically decreased in the patients of MGD [33] Thus, the downregulation of EDA will result in the progress of MGD and the destruction of the cornea and conjunctiva.

### 7.2. Lacrimal Gland

The LG secretes the aqueous layer of the tear film [74]. Although the EDA activity was observed to progressively decrease during development [52], the quantity and quality of tear production by the LG was dramatically alternated in progressive XLHED patients and animal models. The LG weight was increased in EDA^−/−^ mice compared with wild-type mice [52]. Moreover, the terminal differentiation of cells was found to be altered to an unmatured state in all the LG compartments including the epithelium of ducts and acinar, and myoepithelial cells in EDA^−/−^ mice [52]. Indeed, a proper terminal differentiation is crucial for physiological LG secretion [52]. The EDA pathway not only maintains appropriate cell differentiation but also mediates the expression of the protective secretory factors found in the tear film. Additionally, the growth factors and inflammatory cytokines such as growth differentiation factor 5 (Gdf5), C-X-C motif chemokine ligand 10 (CXCL10) known to be secreted in basal tears were downregulated in the EDA^−/−^ LG [52,71]. It is a remarkable fact that Gdf5 is involved in the inhibition of corneal epithelial cells’ proliferation [75], while CXCL10 is associated with dry eye [76]. Moreover, it is shown that EDA^−/−^ animals presented with delayed corneal wound healing [70], which could possibly be due to LG maturation defects, and the TGF-β1, FGF7 and HGF of the lacrimal gland showed abnormal expression during this process. Surprisingly, inhibiting EDA signaling in the LG epithelium seems to be part of a feedback loop between the cornea and LG, which allows the secretion of reflex tears supporting corneal wound healing [52]. Similar to the meibomian gland, the EDA could maintain the homeostasis of the LG and promote tear production to support the cornea and conjunctiva.

### 7.3. Cornea

The corneal epithelium weakly expresses EDA protein, whereas it significantly expresses EDAR [33] (Figure 1). Few researchers have reported that EDA signaling is inactive in the cornea during physiological and pathological conditions [52]. However, the corneal changes, such as corneal defect and keratitis was age dependent in patients and animal models with EDA mutation [52,63,64]. The corneal epithelial integrity was defective and the thickness was reduced in the early postnatal stage of EDA mutant *Tabby* mice, with the decrease in corneal epithelial proliferation and delayed corneal wound healing [33]. EDA-mutated *Tabby* mice also displayed significant inflammation of the ocular surface and corneal pannus during their adult stage [70]. Primarily, these defects were assumed to be induced by the alteration of the tear film lipid layer in MGD and the reduction in tear production by LG dysfunction [64,77]. More recently, researchers have concluded that these syndromes are a primary sign of XLHLED, i.e., EDA deficiency [78]. In our previous study, we found that EDA contributes to the maintenance of the epithelial barrier function [70], with the upregulating of ZO-1 and claudin-1 expression through the activation of the sonic hedgehog signaling pathway [70]. Our study also showed that EDA could promote corneal epithelial cell proliferation through regulation of the EGFR signaling pathway [33]. Exogenous EDA protein could rescue the normal corneal epithelial morphology in the EDA-deficient *Tabby* mice [33]. In our view, although the alternation of the meibomian gland and the LG contributed to the dysfunction of the cornea, EDA expression may directly balance the hemostasis of the corneal epithelium by promoting the proliferation and thereby maintaining the barrier function. The conditional dysfunction of the EDA receptor in the cornea can be attempted to better understand the function of EDA in the cornea.

## 8. Therapeutic Efficiency of Recombinant EDA 

XLHED is a systemic genetic disease caused by mutation of the EDA gene and deficiency of the signaling protein EDA [79,80], which leads to the abnormal development of exocrine glands, hair and teeth [81]. The affected individuals inescapably suffer from severe MGD and a dry eye phenotype [53], along with chronic conjunctivitis and blepharitis [33]. 

In 2003, Olivier Gaide et al. synthesized soluble recombinant fusion forms of EDA, namely Fc: EDA1 and Fc:EDA2, and these were effective when tested on the *Tabby* mice model [68]. Meanwhile, researchers found that prenatal ultrasound-guided intra-amniotic injections [82] or the postnatal intravenous administration of soluble recombinant EDA (Fc: EDA1) [65] can efficiently modify the disease development in the XLHED animal models [65,66]. Additionally, after the Fc: EDA1 treatment during the postnatal period, meibomian gland and eyelid development were successful and tear production was significantly increased. Meanwhile, the rate of keratoconjunctivitis sicca and eye infection incidence was greatly reduced [63,64,80]. Taken together, these results indicated that the soluble recombinant EDA (Fc: EDA1) applied during the postnatal period can efficiently maintain the homeostasis of the ocular surface. Furthermore, in the mEDA-A1 transgenic *Tabby* mice, the meibomian glands were restored considerably. Meanwhile, the neovascularization, keratitis, ulceration and keratinization of the cornea, and blepharitis and conjunctivitis of the ocular surface inflammation were significantly prevented [70]. In our study, we found that applying mouse recombinant EDA protein in the conjunctival sac of *Tabby* mice significantly promoted epithelial wound healing and the proliferation of the corneal epithelium [33]. Thus, recombinant EDA could be a promising therapeutic candidate in the reconstruction of the ocular surface homeostasis. What is more, the TNF ligands and receptor binding with the Fas-associated protein with death domain (FADD) family adaptors or TNF-R-associated factor (TRAF) family adaptors were composed by trimerization. The FADD often mediate apoptosis and TRAF mediate cell differentiation and inflammation. EDA-EDAR through EDARADD recruits tumor necrosis factor TRAFs for the activation of the downstream NF-κB signaling pathway and other signaling pathways. The abnormal expression of EDA may induce inflammation. However, the possible inflammatory involved effect of EDA should be further studied. Although there was no significant side effect by systemic or local allied EDA, the safety of recombinant ectodysplasin A1 replacement protein in human subjects should also be further investigated.

## 9. Perspective

As the EDA-EDAR system governs prenatal development, the application of recombinant EDA protein could be a boon in treating XLHED. Further pharmacological studies and genetic studies should focus on determining the downstream of the EDA signaling pathway to correct ocular surface disorders and the crosstalk between other pathways to promote corneal homeostasis. What is more, exogenous EDA performed a promising therapeutic effect on the ocular surface destruction of EDA mutation-induced MGD, and the local application of EDA in ocular surface destruction, such as MGD and corneal epithelium dysfunction, should be confirmed in the future.

The Initial regulatory steps in the EDA signaling pathway are still not fully understood. In the ocular surface, EDA could upregulate the expression of Ki67, EGFR, p-EGFR and p-ERK in the corneal epithelium. In other words, EDA can promote corneal epithelial cell proliferation through regulation of the EGFR signaling pathway [33] (Figure 2, Table 2). The role of EDA in the crosstalk of the EGFR signaling pathway, the sonic hedgehog signaling pathway and the downstream regulators needs further study. The EDA-EDAR-involved NF-κB signaling pathway, Wnt/β-catenin signaling pathway, BMP signaling pathway and c-Jun N-terminal kinase signaling pathway are highly related to the pathology of the ocular surface, and the correlation of these signaling pathways and EDA-EDAR should be further studied in the ocular surface. During the development of the meibomian gland, EDA targets Lrp6-DKK4 to modulate Wnt action to regulate meibomian gland induction [22] (Table 2). However, the role of EDA in maintaining the homeostasis of the meibomian gland is not well elucidated. 

The majority of the EDA in the tear film is produced by the meibomian gland, and it is dramatically decreased in MGD. Thereby, the detection of EDA in tears could be an index to assess and diagnose the function of the meibomian gland [33]. The correlation of EDA in tears with the grade of MGD needs to be further studied. As we reported that EDA could maintain the barrier function and promote the proliferation of the corneal epithelium, the correlation of corneal epithelium dysfunction and the quality of tear EDA should be further confirmed, especially in persistent corneal epithelium defects. Thus, the potential applications of EDA in ocular surface health and diseases remains to be widely researched.

## Figures and Tables

**Figure 1 ijms-23-15700-f001:**
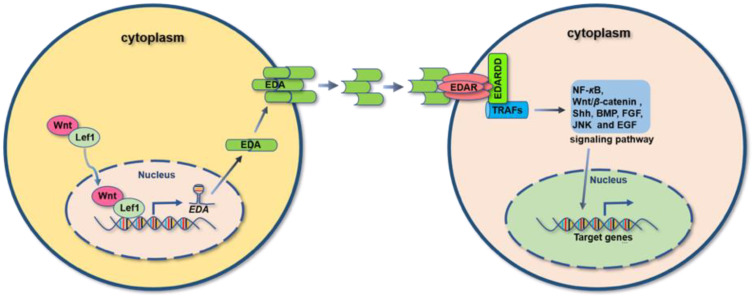
Schematic representation of EDA signaling.

**Figure 2 ijms-23-15700-f002:**
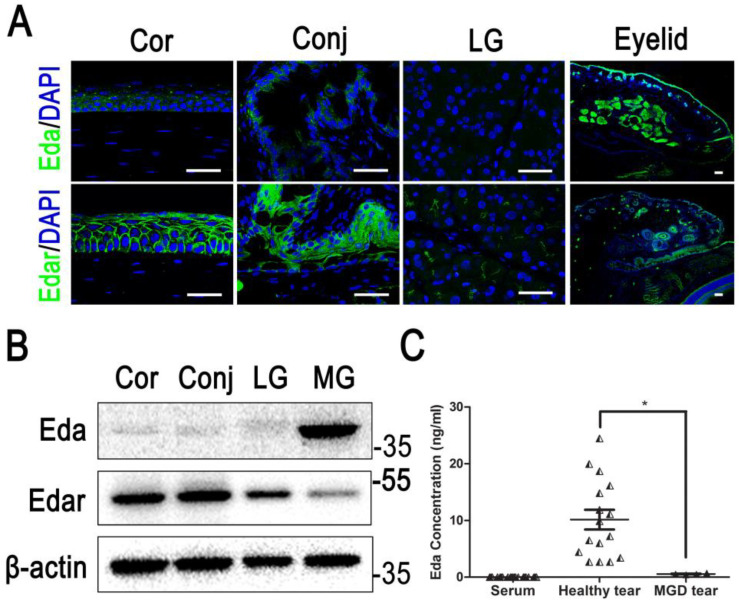
The expression of EDA-EDAR in the ocular surface [33]. (**A**) Immunofluorescent staining of EDA and EDAR expression in the corneal (Cor) and conjunctival (Conj) epithelium, lacrimal gland (LG) and meibomian gland (MG). (Scale bars represent 50 μm). (**B**) Western blot results of EDA and EDAR in corneal epithelium, conjunctiva, lacrimal gland and meibomian gland tissues. (**C**) ELISA results of EDA in normal human serum, healthy human tear and tears from MGD patients (* *p* < 0.05). (Reproduced from Ref. [33]).

**Figure 3 ijms-23-15700-f003:**
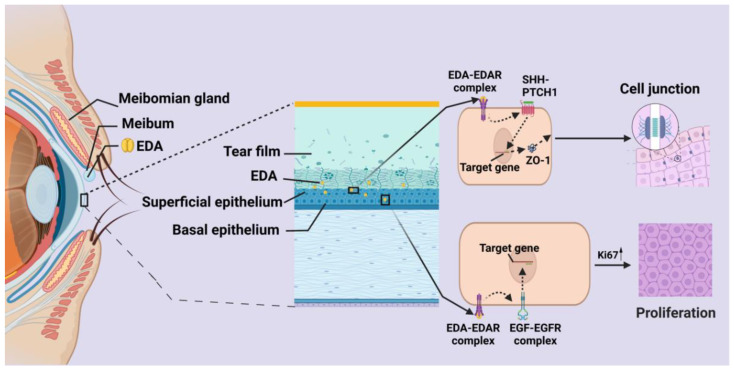
Schematic representation of EDA in ocular surface homeostasis. In ocular surface, the EDA was mainly contributed by meibomian gland. EDA plays a crucial role in maintaining the ocular surface hemostasis by promoting proliferation though EGF signaling pathway and maintaining normal barrier function of corneal epithelium by SHH signaling pathway.

**Table 1 ijms-23-15700-t001:** Function of EDA in organs and tissues.

	Physiology	Pathology
**Heart**	Expression [11]	NA
**Kidey**	Expression in kidney epithelial cells [37]	Polycystic kidney dysplasia [38], diabetic kidneys [37].
**Pancreas**	Expression [39,40]	Insulin resistance [39]
**Brain**	Expression [41]	NA
**Lung**	Expression in distal tracheal regions and the distal lung [42]. The development of submucosal glands [43].	Lung infection [44,45], high prevalence of asthma-like symptoms [46]
**Liver**	Expression in hepatic stellate cells [28,47]	Increases in non-alcoholic fatty liver disease and insulin resistance [28]
**Skeletal muscle**	Expression in muscle cells [28]	Insulin resistance [28]
**Skin and skin appendages**	Expression in epithelium. The formation of skin appendages [35] and skin repair [48]	Defective formation and further morphogenesis dysfunction of hair follicles, sweat glands and teeth [35], delay in healing [48]
**Mammary glands**	Expression in mammary epithelium. Mammary placode formation and branching morphogenesis [31,49,50]	Smaller ductal trees [16,51]
**Ocular surface**	Expression in meibomian gland epithelium. Meibomian gland formation [19] and lacrimal gland morphogenesis [52]	Dry eye [53], delay in healing [33]

NA, no study.

**Table 2 ijms-23-15700-t002:** Expression of EDA and pathways involved in ocular surface homeostasis.

	EDA	EDAR	Function	Pathway	Pathology
**Cornea**	+	+++	Proliferation Barrier function	EGF [33] SHH [70] ERK [33]	Corneal defect, keratitis, decrease in corneal epithelial proliferation and delayed corneal wound healing [33]
**Conjunctiva**	+	++	NA	NA	
**Meibomian gland**	+++	++	Development	WNT [22]	Abnormal development of the meibomian gland [33]
**Lacrimal gland**	+/−	++	Development Lacrimation	Cxcl10 [52,71]	The terminal differentiation of cells was abnormal and a decrease in tear production [52]

+, weak expression; +++, high expression; ++, between + and +++; +/−, no confirmed; NA, no study.

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
