# Peer review of "The Role of Ectodysplasin A on the Ocular Surface Homeostasis"

_ijms, 2022, doi:10.3390/ijms232415700_

Round 1

Reviewer 1 Report

This review article is well written. The absorption of exogenous EDA on the ocular surface required further investigation for it's clinical use.

Author Response

     Thanks for the reviewer for the positive comments on our manuscript. In our future clinical study, we will investigate the absorption of exogenous EDA on the ocular surface. In our previous animal study, we did not investigate this. Alternatively, we observed the effect of EDA on epithelial cell junction formation and corneal epithelial cell proliferation using in vivo animal models.

Reviewer 2 Report

Ou et al. provide a summary of the physiologic and pathologic role of ectodysplasin A on the ocular surface. While the review is logically organized, it is predominantly based on significantly older studies. Recent studies on this topic are from the reviewers' own work, which diminishes the value of the review. 

Minor points:

- there are numerous typographical errors throughout the manuscript that makes it very difficult to read and understand.

- Table 1 could be improved by specifying the cell types where EDA is expressed under "physiology"

- there are multiple references to the roles of the LG, MG and tear film that are not as accurate as they should be. For example, under 5. The homeostasis of ocular surface, the last sentence should be corrected to "the meibum, aqueous, and mucin mix to form the tear film..." The manuscript should be revised to address these issues.

- it would be helpful if the authors provide a hypothesis as to why there is an inverse relationship between the expression of Eda and its receptors in the ocular tissues.

- Figure 3 has numerous spelling errors. It should also be more anatomically accurate with the actual tissues depicted in their relative position to the eye and the conjunctiva and cornea labeled.

- It is unclear why Fig2 and 3 are referenced at the end of the last sentence on pg 6. Neither figures show the pathologic results from the downregulation of EDA on the ocular surface.

- A figure that illustrates the potential signaling pathways involved in the eye would be helpful.

Author Response

Thanks for the reviewer’s comments. As per the reviewer’s suggestion, proof reading was performed on this manuscript. We did thorough literature research and all the reports related to the role of EDA on the ocular surface were reviewed and discussed in our manuscript. We wish our review is comprehensive and provide a prospective view for future study in this field.

- Table 1 could be improved by specifying the cell types where EDA is expressed under "physiology"

Response:

The reviewer’s comments are well accepted. As per the reviewer’s suggestion we have updated the Table 1.

- there are multiple references to the roles of the LG, MG and tear film that are not as accurate as they should be. For example, under 5. The homeostasis of ocular surface, the last sentence should be corrected to "the meibum, aqueous, and mucin mix to form the tear film..." The manuscript should be revised to address these issues.

Response:

The reviewer’s comments are well accepted. As per the reviewer’s suggestion the references and the incorrect sentence was modified in the manuscript.

- it would be helpful if the authors provide a hypothesis as to why there is an inverse relationship between the expression of Eda and its receptors in the ocular tissues.

Response:

We have discussed it in the manuscript. The avascular cornea is nourished by the blood vessels of the limbus, the aqueous humor and tear. Our study for the first time found that the cornea expressed receptors of EDA while the EDA was produced by the Meibomian gland.

- Figure 3 has numerous spelling errors. It should also be more anatomically accurate with the actual tissues depicted in their relative position to the eye and the conjunctiva and cornea labeled.

Response:

Thanks for pointing out this. We have updated the Figure 3.

- It is unclear why Fig2 and 3 are referenced at the end of the last sentence on pg 6. Neither figures show the pathologic results from the downregulation of EDA on the ocular surface.

Response:

The authors cited Fig2 and 3 at the end of the paragraph the pathologic results from the EDA expression on the ocular surface are concluded in Table 2.

- A figure that illustrates the potential signaling pathways involved in the eye would be helpful.

Response:

We have added the involved signaling pathways in Table 2.

Reviewer 3 Report

This manuscript reviewed the role of ectodysplasin A (EDA), a member of the tumor necrosis factor (TNF) superfamily, on ocular surface homeostasis. This review discussed the EDA and EDAR signal system and its function in the physiology and pathology of ocular surface development. Also, the potential therapeutic effects of EDA were discussed based on animal studies.

Please discuss more on the details of the EDA-EDAR signal pathway, and its distinct role in the members of the TNF superfamily.

Author Response

Please discuss more on the details of the EDA-EDAR signal pathway, and its distinct role in the members of the TNF superfamily.

Response:

Thanks for the reviewer’s constructive suggestion. We have added details about the EDA-EDAR signal pathway in the manuscript.

Reviewer 4 Report

In table 1 and 2:  some rows overlap and cannot be read well.

Author Response

In table 1 and 2: some rows overlap and cannot be read well.

Response:

We thank the reviewer for pointing out the error. We have modified the lay out of the tables

Reviewer 5 Report

The authors should describe the pathophysiology of dry eye syndrome to highlight the role of EDA on ocular surface. Discuss about the EDA functions for each mechanism of dry eye.

My PDF file presents "Numerous studies with EDA mutant Tabby mice have proved that EDA-EDAR regulates the NF- B signaling pathway." in the second page. It should be revised.

Figure 3 - No EDA and its roles in the figure.

Can the authors discuss about the safety of recombinant ectodysplasin A1 replacement protein in human subjects?

Is it possible to use as an anti-inflammatory therapy for ocular inflammatory diseases or dry diseases?

Author Response

The authors should describe the pathophysiology of dry eye syndrome to highlight the role of EDA on ocular surface. Discuss about the EDA functions for each mechanism of dry eye.

Response:

We thank the reviewer for the constructive comments. We elaborated the possible mechanism of the effect of EDA on dry eye in the modified manuscript.

My PDF file presents "Numerous studies with EDA mutant Tabby mice have proved that EDA-EDAR regulates the NF-kB signaling pathway." in the second page. It should be revised.

Response:

We have revised this sentence into: “Many studies on EDA mutant Tabby mice have proved that EDA-EDAR regulates the NF-kB signaling pathway”.

Figure 3 - No EDA and its roles in the figure.

Response:

The reviewer’s comments are well accepted. We have updated Figure 3 and added the roles of EDA on the ocular surface in the figure so that it’s more readable.

Can the authors discuss about the safety of recombinant ectodysplasin A1 replacement protein in human subjects?

Response:

We thank the reviewer for the critical comment.

As reported by Hermes, K., et al. (Hermes, K., et al., Prenatal therapy in developmental disorders: drug targeting via intra-amniotic injection to treat X-linked hypohidrotic ectodermal dysplasia. J Invest Dermatol, 2014. 134(12): p. 2985-2987.) , Casal, M.L., et al.( Casal, M.L., et al., Significant correction of disease after postnatal administration of recombinant ectodysplasin A in canine X-linked ectodermal dysplasia. Am J Hum Genet, 2007. 81(5): p. 1050-6.) and Margolis, C.A., et al (Margolis, C.A., et al., Prenatal Treatment of X-Linked Hypohidrotic Ectodermal Dysplasia using Recombinant Ectodysplasin in a Canine Model. J Pharmacol Exp Ther, 2019. 370(3): p. 806-813.) that prenatal ultrasound-guided intra-amniotic injections or postnatal intravenous administration of soluble recombinant EDA (Fc: EDA1) can efficiently modify the disease development in the XLHED animal models. In our previous study, we also found that locally applying EDA could improve the corneal pathological changes in Tabby mice. So, we concluded that recombinant ectodysplasin A1 replacement protein in human subjects should be safe. However, future studies are required in the field.

Is it possible to use as an anti-inflammatory therapy for ocular inflammatory diseases or dry diseases?

Response:

We thank the reviewer for the thoughtful comment. As reported by Qingjun Zhou that the inflammation related signal transducer and activator of transcription-3 (STAT3) was activated on the cornea and conjunctiva of Tabby mice. However, the possible anti-inflammatory effect of EDA on the ocular surface is worthy of further study.

Round 2

Reviewer 5 Report

What a good review article. Acceptable.